# Investigating gene-environment interaction on attention in a double-hit model for Autism Spectrum Disorder

**Melvin Maroon**[1], **Faraj Haddad**[1], **Ella Doornaert**[1], **Brian Allman**[1,2], **Susanne Schmid**[1,2,3¤]*

**1** Neuroscience Graduate Program, The University of Western Ontario, London, ON, Canada, **2** Anatomy & Cell Biology, Schulich School of Medicine & Dentistry, The University of Western Ontario, London, ON, Canada, **3** Psychology, The University of Western Ontario, London, ON, Canada

¤ Current address: Schulich School of Medicine & Dentistry, The University of Western Ontario, London, ON, Canada
* Susanne.schmid@schulich.uwo.ca

**Data Availability Statement:** All data are displayed in the paper or supplemental data. Raw data of the touch screen task are freely available on figshare, DOI: 10.6084/m9.figshare.25408720.

## Abstract

Autism Spectrum Disorder (ASD) is a neurodevelopmental behavioral disorder characterized by social, communicative, and motor deficits. There is no single etiological cause for ASD, rather, there are various genetic and environmental factors that increase the risk for ASD. It is thought that some of these factors influence the same underlying neural mechanisms, and that an interplay of both genetic and environmental factors would better explain the pathogenesis of ASD. To better appreciate the influence of genetic-environment interaction on ASD-related behaviours, rats lacking a functional copy of the ASD-linked gene *Cntnap2* were exposed to maternal immune activation (MIA) during pregnancy and assessed in adolescence and adulthood. We hypothesized that *Cntnap2* deficiency interacts with poly I:C MIA to aggravate ASD-like symptoms in the offspring. In this double-hit model, we assessed attention, a core deficit in ASD due to prefrontal cortical dysfunction. We employed a well-established attentional paradigm known as the 5-choice serial reaction time task (5CSRTT). *Cntnap2*$^{-/-}$ rats exhibited greater perseverative responses which is indicative of repetitive behaviors. Additionally, rats exposed to poly I:C MIA exhibited premature responses, a marker of impulsivity. The rats exposed to both the genetic and environmental challenge displayed an increase in impulsive activity; however, this response was only elicited in the presence of an auditory distractor. This implies that exacerbated symptomatology in the double-hit model may situation-dependent and not generally expressed.

## Introduction

Autism Spectrum Disorder (ASD) is a neurodevelopmental behavioral disorder affecting approximately one in 100 children worldwide [1]. The Diagnostic and Statistical Manual of Mental Disorders (DSM-5) criterion for ASD focuses on deficits in social behavior and communication, as well as restricted interests and repetitive actions [2]. Intellectual disabilities and abnormal processing of sensory input are also seen in individuals with ASD [2]. Accordingly,

**Funding:** This study was funded by CIHR to SS. The funders had no role in study design, data collection and analysis, decision to publish, or preparation of the manuscript.

**Competing interests:** The authors have declared that no competing interests exist.

past research has reliably shown that individuals with ASD demonstrate difficulty accurately perceiving social cues and exhibit inflexible behavior [3–5]. Inflexible behavior is commonly seen through repetitive motor actions and strict custom routines [6, 7]. Presently, ASD is predominantly diagnosed through behavioral assessment [2]. Due to the number of potential risk factors, both genetic and environmental, it is valuable to investigate their roles in the emergence of ASD-like symptomatology.

The emergence of genetic sequencing technology has helped researchers elucidate the bases of neurobiological disorders that were previously categorized as idiopathic [8]. Through large-scale sequencing cohorts, researchers have discovered that the genetic basis of ASD is widely heterogenous. ASD can be caused by single-gene (monogenic) mutations. For instance, a mutation in the X-linked FMR1 gene, commonly known as Fragile X Syndrome, accounts for approximately 2% of identified cases [9]. Contrarily, rare and common variants are not necessarily pathogenic [10, 11]. The impact of such risk factors is dependent on the type of variant, the modified genes, and the compounded effect of additional genetic and environmental impacts [12, 13]. For instance, a large-scale genome-wide association study discovered 102 common variants associated with ASD; however, they only account for around 3.5% of the total heritability of the disorder [14].

Beyond genetics, many sources discuss the influence of environmental factors on the etiological basis of ASD; these include parental age, maternal medication use, and post-natal family environment [15, 16]. More recently, the maternal inflammatory response, triggered by a viral infection, has been found to harmfully affect fetal development [17]. While certain rare pathogens pose distinctive threats to the brain, the infection caused by several viruses result in a similar risk for ASD development. This implies that the induced maternal immune activation (MIA) is the core mechanism impacting the fetus, as opposed to the virus itself [18–20]. The MIA hypothesis has been validated by activating the immune system with a range of pathogens during pregnancy. Subsequently, the offspring are assessed for corresponding symptomatology that is attributed to human neurodevelopmental disorders [19, 21]. A greater understanding of the effects of maternal immune reaction on fetal brain development is imperative, especially with the ongoing after-effects of the COVID-19 pandemic.

With the growing repertoire of genetic and environmental risk factors attributed to ASD, more studies have begun characterizing double-hit gene-environment models for neurodevelopmental disorders [22, 23]. For example, a national surveying study in Sweden revealed that ASD heritability is at approximately 50%, reinforcing the proportionally impactful role of the environment [24]. Additionally, in a study centered around maternal metabolic and inflammatory conditions, children with diabetic mothers had an increased risk of ASD onset compared to children with a genetic predisposition alone [25]. Nonetheless, each human is genetically and behaviorally dissimilar, undermining potential risk-factor interactions [26]. Animal models alone can substantiate the claim that environmental risk factors exacerbate genetic predispositions [26]. The present study aimed to characterize a double-hit gene-environment rodent model for ASD, particularly focusing on the process of attention.

The genetic factor in our present model is the loss-of-function of the Contactin-associated protein-like 2 (*CNTNAP2*) gene. *CNTNAP2* is highly expressed in brain areas implicated in ASD and poses a risk through both common and rare variations [27, 28]. *CNTNAP2* codes for a cell adhesion molecule involved in synaptic formation, cortex organization and neuronal migratory function [29, 30]. Consequences of *CNTNAP2* loss-of-function mutations were uncovered by Strauss et al. [31] in a population of Old Order Amish children. Out of the affected population, 70% exhibited ASD-associated symptoms [31]. The preclinical Sprague-Dawley rodent model with a homozygous gene knockout (*Cntnap2* KO) displays endophenotypes including hyperactivity, repetitive behaviors, and reduced vocalizations [32, 33]. For the

environmental challenge, we used polyinosinic polycytidylic acid (poly I:C) to trigger an anti-viral-like immune response in pregnant dams. Poly I:C elicits its effects through toll-like 3 receptors (TLR3), a highly conserved innate immune receptor [34, 35]. Due to the proven efficacy of the Sprague-Dawley poly I:C model, we studied the effect on attention, an ASD related behavior, in combination with the *Cntnap2* risk factor.

Visuospatial attention is an integral characteristic when assessing social-communicative deficits attributed to ASD [36]. Selective attention and orienting have a critical role in cognitive development and can even play a role in regulating emotional states in humans [37]. Prior studies have found that children with ASD have lessened attention, particularly to facial cues; however, most rodent models focus on non-salient visual cues [38, 39]. Based on previous reports on attentional deficits in ASD, we hypothesized that *Cntnap2* deficiency interacts with poly I:C immune activation to exacerbate ASD related attentional changes exhibited by each model alone. We employed the 5-choice serial reaction time task (5CSRTT), a standard rodent behavioral task used to assess attention [40]. The 5-CSRTT was developed as the rodent equivalent to Leonard's choice reaction time task in the Cambridge Neuropsychological Test Automated Battery [41]. We predicted that the double-hit model displays poorer performance on the 5CSRT task compared to either single-hit model.

## Materials and methods

### Animals

The study was conducted with 19 wildtype (WT) and 22 homozygous knockout (*Cntnap2* KO) male Sprague-Dawley rats. The n-value may differ in the displayed test results as certain rats were seizure prone and exhibited atypical baseline behavior. Horizon Discovery (Boyertown, PA; originated at SAGE Laboratories, Inc. with Autism Speaks; the line is presently upheld by Envigo) provided heterozygous breeders with a five base-pair deletion at exon six in the *Cntnap2* gene, created by zinc-finger nuclease target site CAGCATTTCCGCACC|aatgga| GAGTTTGACTACCTG. All genotypes tested in this experiment were littermates acquired from heterozygous crossings.

Pregnant dams were exposed to either saline or poly I:C on gestation day (GD) 9.5. This timepoint corresponds to the first trimester in humans, where the risk of MIA on ASD onset in the offspring is highest [42]. On GD 9.5, pregnant females (n = 16; $n_{poly I:C}$ = 8, $n_{saline}$ = 9) underwent brief isoflurane anesthesia. The dams were injected with either 0.9% saline or 4 mg/kg poly I:C (Sigma Lot#037M4011V) into the tail vein. From weaning (PD 21), the offspring were placed in open cages, provided with ad libitum food and water, and were put on a 12-hour light– 12-hour dark cycle. Polycarbonate huts and crinkled paper supplemented the cage environment. The offspring were housed in same-sex pairs prior to and during behavioral testing. All behavioral testing was conducted in the light-cycle (7:00 a.m. to 7:00 p.m.).

### 5-choice serial reaction time task (5-CSRTT)

At PD 100, the rats were food restricted to a target weight of 90% of their normal weight. Through operant conditioning, the rats were trained to locate and report a passing visual stimulus shown pseudo-randomly in one of five sites on a horizontal mask of apertures. The detailed protocol for pretraining and baseline acquisition are well documented in a priorly published protocol by Mar and colleagues [43].

**Task outline for 5-CSRTT protocol.** The default house-light setting in the chamber is off for all training and testing sessions. At the beginning of every session, a pellet (50 food: 50 sugar) is dropped in the food tray to motivate the rat to begin the task. To initiate the trial, the rat must nose poke inside the food tray and activate the sensor. There is a 5 s delay prior to

stimulus presentation. The rat must then nose poke the illuminated stimulus panel to collect a food reward at the tray. If the correct panel is selected, a pellet is dropped with a pure tone. If an incorrect panel is selected, the house light is turned on with a white tone as punishment. After a 5 s intertrial interval, the tray illuminates to instigate the start of the subsequent trial. The stimulus duration is dependent on the training or testing stage. When the rat does not interact with the screen during stimulus presentation, the trial is considered an omission. If the rat nose pokes the screen during the delay period, the trial is labelled as a premature response. Lastly, if the rat continuously nose pokes the panel after a correct choice, its is labelled as a perseverative response.

**Pretraining and baseline training period.** Pretraining allowed the rat to acclimate to the touchscreen chamber and screen. The rats also learned the fundamentals of the system; this includes habituation, initiating the task, and associating a correct response with a sugar-pellet reward [43]. The training period was divided across 13 baseline levels. The session length for all levels was set at 60 minutes, with a total of 60 trials per session. The intertrial interval was set at five seconds across all training stages. The latency or delay prior to the stimulus was also set at five seconds across all training stages. The stimulus duration was set at 60 seconds and was gradually decreased to 0.5 seconds at Baseline Stage 13. To move past a stage, the rat had to achieve greater than 80% accuracy, and omit less than 20% of trials. The results display the findings from Baseline 13 (0.5 second delay) as it corresponds to the duration value used in the test paradigms.

This study employed a fixed training period structure. Once the rats completed pretraining, they had 40 days to complete the training phase prior to testing. If the rat completed the training in less than 40 days, the testing paradigms were introduced earlier. This structure ensured that the rats showing no improvement did not overtrain at lower baseline levels indefinitely. Additionally, this structure allowed fast learners to complete the training phase without being overtrained, which is a limitation that can be seen in a group training structure where all rats would be trained until the very last rat completed their baseline training. Rats unable to complete baseline training independently were moved to the final baseline training stage on day 40. Extreme outliers were removed prior to the initiation of the testing protocol.

**Test paradigms.** The first test was a Short-Delay variation. This test randomized the delay prior to stimulus onset at values less than the standard 5-second delay. This variation assessed global attention, providing the rat with less time to prepare for stimulus onset. The second test paradigm was a Long-Delay variation. The delays prior to stimulus onset were set at values greater than the standard 5-second delay. This test assessed inhibition and impulsivity because the rat was provided more opportunity to prematurely respond prior to stimulus onset. The third test paradigm was the Distraction variation. This test employed an auditory distractor–noise of 105dB at randomized time points between initiation and stimulus presentation. This test measured selective attention. Each of the four test paradigms were set at 100 trials to be completed within 60 minutes or less.

**Measures for attention processing.** The following measures were used to assess the rats' executive function: response accuracy (number of correct over all trials) to measure of attentional selectivity, omissions (trials with no response) to measure sustained attention, and premature responses (nose poke prior to stimulus) or perseverative correct responses (continued response after correct action feedback) as measures of impulsivity and compulsion, respectively.

## Statistics

All data are mean ± standard deviations unless otherwise stated. Data was analyzed using a two-way analysis of variance ANOVA for the baseline task acquisition stage. The two

between-subjects factors for the two-way ANOVA were genotype (WT, *Cntnap2* KO) and prenatal exposure (saline, poly I:C). A three-way mixed analysis of variance ANOVA (two between-subject factors and one within-subjects factor) was used for the three test paradigms: Short Delay, Long Delay, and Distraction. The three-way ANOVA was followed by the Bonferroni pairwise comparison post hoc, utilizing the IBM SPSS statistics software. The statistical significance was placed at $p < .05$.

## Results

### 5CSRTT training: Baseline stage 13 (0.5s stimulus)

Animals needed between 30 and 40 days of training to reach testing criterion independently of experimental group (see S1 Fig in S1 File). A two-way ANOVA was conducted to examine the effects of genotype (*Cntnap2* $^{+/+}$ [*Cntnap2* WT] or *Cntnap2* $^{-/-}$ [*Cntnap2* KO]) and prenatal exposure (saline or poly I:C [MIA]) on each measure of attention for the last baseline training stage (baseline 13). During training, average accuracy on baseline performance was used to assess attentional selectivity. Neither genotype nor prenatal exposure significantly influenced performance accuracy for baseline training (genotype main effect: $F(1, 42) = 0.338$, $p = 0.564$, partial $\eta2 = 0.009$; prenatal exposure main effect: $F(1, 42) = 3.550$, $p = 0.067$, partial $\eta2 = 0.085$; **Fig 1A**). There was also no significant interaction effect between genotype and prenatal exposure on average accuracy for baseline stage 13 ($F(1, 42) = 0.420$, $p = 0.521$, partial $\eta2 = 0.011$). The measure of omission was used to assess sustained attention during task acquisition. There was no main effect seen by either risk factor alone (genotype main effect: $F(1, 42) = 2.489$, $p = 0.123$, partial $\eta2 = 0.061$; prenatal exposure main effect: $F(1, 42) = 0.037$, $p = 0.849$, partial $\eta2 = 0.001$; **Fig 1B**). No significant interaction between the models was seen ($F(1, 42) = 0.238$, $p = 0.628$, partial $\eta2 = 0.006$).

Premature responses, which were used to measure motor impulsivity, did not reveal any main effect for either model type (genotype main effect: $F(1, 42) = 0.013$, $p = 0.911$, partial $\eta2 = 0.000$; prenatal exposure main effect: $F(1, 42) = 1.698$, $p = 0.200$, partial $\eta2 = 0.043$; **Fig 1C**), nor a significant interaction between genotype and prenatal exposure ($F(1, 42) = 0.361$, $p = 0.552$, partial $\eta2 = 0.009$). However, *Cntnap2* KO rats exhibited greater perseverative correct responses, a measure of repetitive behaviors, regardless of prenatal exposure (genotype main effect: $F(1, 42) = 6,236$, $p = 0.017$, partial $\eta2 = 0.141$; prenatal exposure main effect: $F(1, 42) = 1.350$, $p = 0.252$, partial $\eta2 = 0.034$; **Fig 1D**; interaction effect: $F(1, 42) = 0.180$, $p = 0.674$, partial $\eta2 = 0.005$).

5CSRTT baseline training revealed a significant increase of perseveration in *Cntnap2* KO rats, but no significant differences in the other measures. This indicates that neither *Cntnap2* KO, nor MIA, has a severe impact on learning in the 5CSRTT task.

### 5CSRTT testing: Short-delay paradigm

The Short-Delay paradigm altered the onset of the stimulus presentation after trial initiation. Specifically, the stimulus was presented either 0.5, 1.5, 3.0, or 4.5 seconds after trial initiation pseudo randomly; the standard delay/latency period on all baseline stages was a 5.0 second delay.

A three-way mixed ANOVA was run to investigate the effects of genotype, prenatal exposure and short-delay value on each respective measure of attention. Rats prenatally exposed to poly I:C exhibited poorer accuracy at the 0.5 second delay, regardless of genotype (prenatal exposure* 0.5 timepoint simple-main effect: $F(3, 111) = 4.681$, $p = 0.044$; genotype main effect: $F(1, 37) = 0.970$, $p = 0.331$; **Fig 2A**). *Cntnap2* KO rats exhibited a lower rate of omissions regardless of prenatal exposure (genotype main effect: $F(1, 37) = 13.610$, $p = 0.001$; prenatal

# Training - Baseline 13

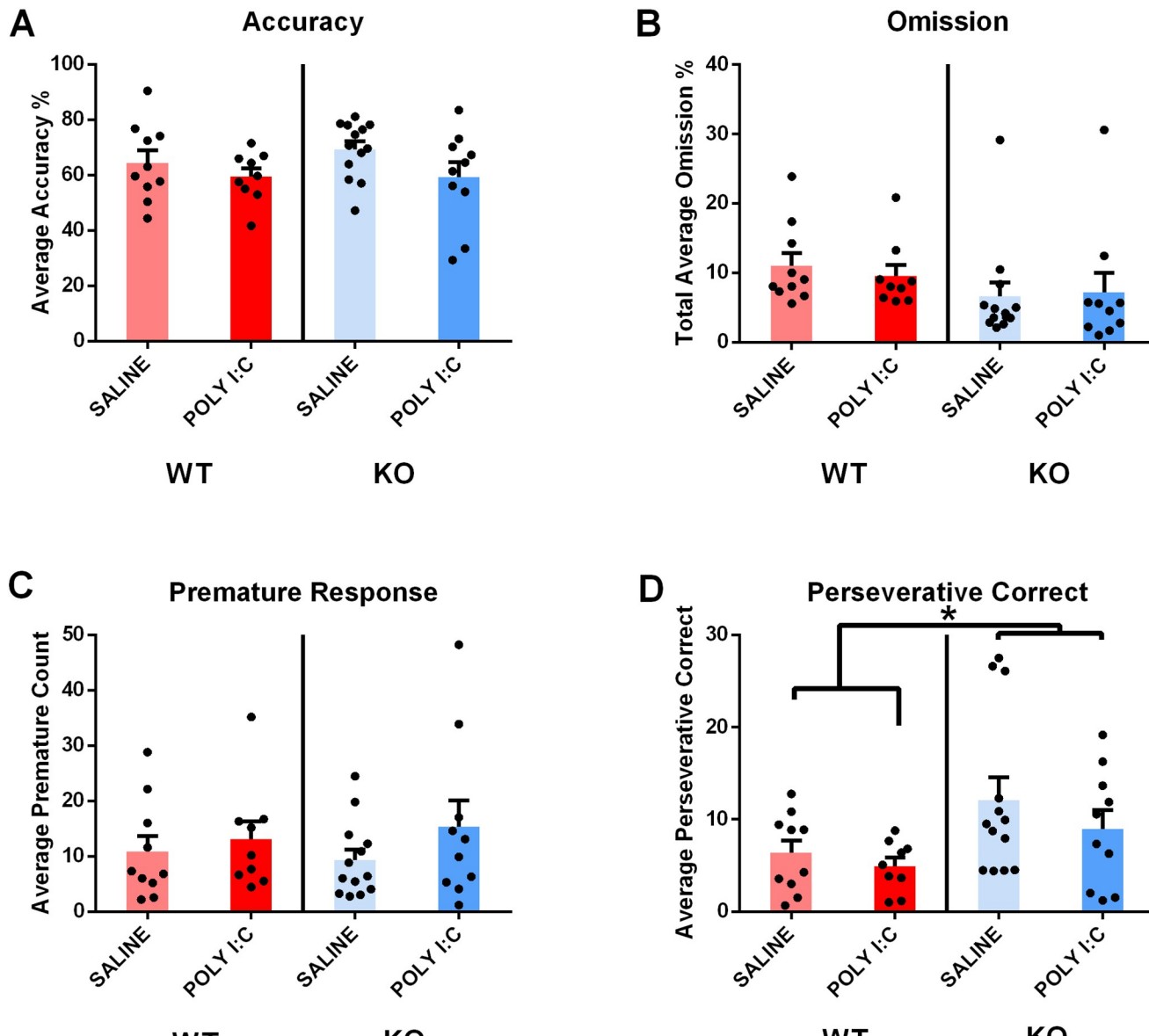

**Fig 1.** A) During the last stage of baseline training, the rats did not exhibit model-dependent accuracy changes; B) changes in omission; C) or significant differences in premature count. D) *Cntnap*2 KO rats exhibited increased perseverative correct responses regardless of poly I:C MIA. *p < .05. Results are shown as mean ± SEM. $N_{WT/SALINE}$ = 10; $N_{WT/POLY\ I:C}$ = 9; $N_{KO/SALINE}$ = 13; $N_{KO/POLY\ I:C}$ = 10.

exposure main effect: F(1, 37) = 0.529, p = 0.472). There was a three-way interaction effect between genotype and prenatal exposure across the delay levels (F(3, 111) = 5.842, p = 0.001). There was a significant two-way interaction between delay and genotype, but not delay and injection (delay*genotype: F(3, 111) = 6.461, p = 0.000; delay*injection: F(1, 37) = 0.873, p = 0.457). Furthermore, there was a two-way interaction between genotype and injection (genotype*injection main effect: F(1, 37) = 5.168, p = 0.029; **Fig 2B**).). All bonferroni-adjusted pairwise comparisons were performed for statistically significant simple simple main effects.

# Testing - Short Delay

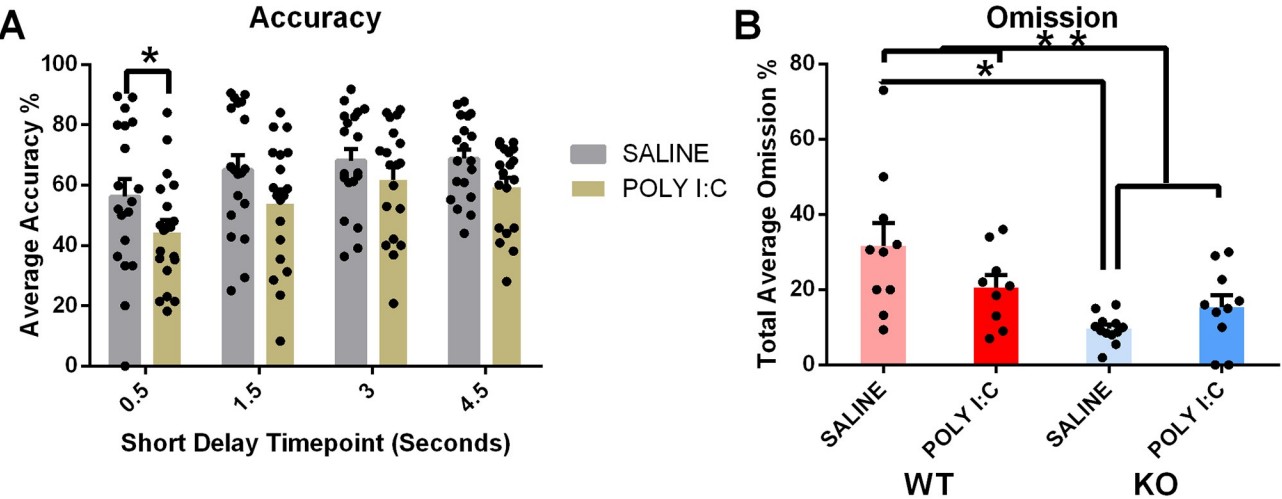

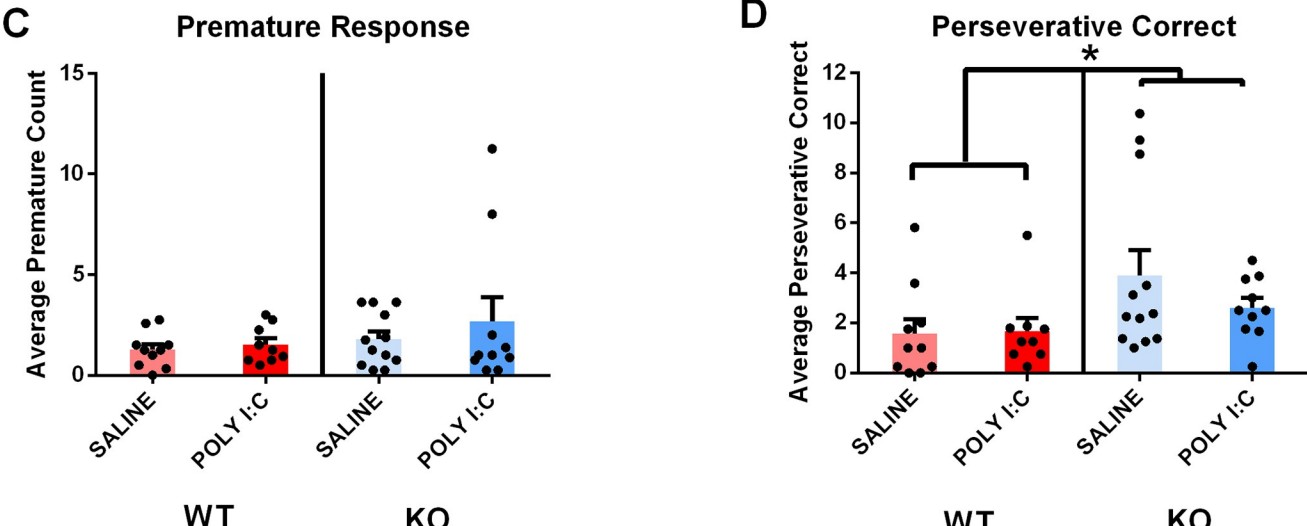

**Fig 2. Short-delay 5CSRTT testing.** A) There was a decrease in accuracy in offspring of poly I:C injected rats at the shortest delay of 0.5 s, regardless of genotype. B) *Cntnap2* KO rats exhibited a decreased omission rate compared to wildtype. There was no simple main effect of genotype on rats exposed to poly I:C; however, there was a simple main effect of genotype on non-exposed rats. C) There was no significant effect of genotype or prenatal exposure, nor any interaction, on premature response. D) *Cntnap2* KO rats exhibited increased perseverative correct responses regardless of prenatal exposure. *p < .05. Results are shown as mean ± SEM. $N_{WT/SALINE}$ = 10; $N_{WT/POLY\ I:C}$ = 9; $N_{KO/SALINE}$ = 12; $N_{KO/POLY\ I:C}$ = 10. For detailed data on different short delay intervals, please see S2 Fig in S1 File.

Saline-exposed Cntnap2 KO rats exhibited a lower rate of omissions than saline-exposed wild-types across all four delays (delay timepoint 0.5 seconds: p < 0.001; delay timepoint 1.5 seconds: p < 0.001; delay timepoint 3.0 seconds: p = 0.008; delay timepoint 4.5 seconds: p = 0.035). Contrarily, prenatally exposed Cntnap2 KO rats did not exhibit a lower omission rate than prenatally exposed wildtypes across all delays (delay timepoint 0.5 seconds: p = 0.468; delay timepoint 1.5 seconds: p = 0.479; delay timepoint 3.0 seconds: p = 0.272; delay

timepoint 4.5 seconds: p = 0.339; S2A Fig in S1 File). For a summary of all detailed statistical results per different delays or distraction parameters, please see S1 Table in S1 File.

The analysis of premature responses did not reveal any main effect in either model type (genotype main effect: F(1, 37) = 1.629, p = 0.210; prenatal exposure main effect: F(1, 37) = 0.747, p = 0.393; **Fig 2C**), nor any significant three-way interaction between genotype and prenatal exposure across the delay levels(F(3, 111) = 0.141, p = 0.935). However, like in Baseline Training, *Cntnap2* KO rats exhibited greater perseverative correct responses regardless of prenatal exposure (genotype main effect: F(1, 37) = 4.929, p = 0.033; prenatal exposure main effect: F(1, 37) = 0.612, p = 0.439; interaction effect: F(3, 111) = 0.562, p = 0.641; **Fig 2D,** for detailed data per delay, please see S2 Fig in S1 File).

Short Delay 5CSRTT testing revealed a deficiency in accuracy in MIA offspring at the shortest delay and confirmed higher perseverative responses in *Cntnap2* KO animals–as also observed in the final baseline training session. Furthermore, there was a lower omission rate in *Cntnap2* Ko animals that was observed in saline animals, but not in Poly I:C injected animals.

## 5CSRTT testing: Long-delay paradigm

The Long-Delay paradigm altered the onset of the stimulus presentation after trial initiation. Specifically, the stimulus was presented either 4.5, 6, 7, or 9.5 seconds after trial initiation in a pseudo random fashion. During Long-Delay testing, neither genotype nor prenatal exposure significantly influenced performance accuracy (genotype main effect: F(1, 37) = 0 .053, p = 0.820; prenatal exposure main effect: F(1, 37) = 2.077, p = 0.158; **Fig 3A**). Also, there was no significant interaction effect between genotype, prenatal exposure and long-delay (F(3, 111) = 0.400, p = 0.753). Likewise, there was no main effect on the measure of omission by either risk factor alone (genotype main effect: F(1, 37) = 1.330, p = 0.256; prenatal exposure main effect: F(1, 37) = 0.036, p = 0.851; **Fig 3B**) nor any interaction between both risk factors (F(3, 111) = 0.192, p = 0.902).

No main effects of genotype or prenatal exposure on perseverative responses were found, nor was there a three-way interaction (genotype main effect: F(1, 37) = 2.755, p = 0.105; prenatal exposure main effect: F(1, 37) = 0.343, p = 0.561; interaction effect: F(3, 111) = 0.433, p = 0.730; **Fig 3C**). There was no main effect on premature responses in either model type alone (genotype main effect: F(1, 37) = 2.217, p = 0.14; prenatal exposure main effect: F(1, 37) = 3.714, p = 0.062), however, there was a significant interaction between genotype and prenatal exposure regardless of timepoint (F(1, 37) = 4.911, p = 0.033). Specifically, WT rats prenatally exposed to poly I:C did not exhibit increased premature responses compared to the WT saline group; however, *Cntnap2* KO rats prenatally exposed to poly I:C exhibited significantly increased premature responses compared to their saline-exposed counterparts (prenatal exposure simple main effect: mean difference [9.051 +/- 3.575], p = 0.016; Fig 3D and 3E). Additionally, *Cntnap2* KO rats prenatally exposed to poly I:C showed higher premature responses in comparison to WT rats prenatally exposed to poly I:C (genotype simple-main effect: mean difference 10.123 +/- 3.331, p = 0.004; **Fig 3F**).

The Long Delay 5CSRTT revealed an interaction of genotype and prenatal exposure, with significantly increased premature responses in poly I:C offspring with *Cntnap2* KO, indicating an accumulating effect of a genetic deletion and MIA on this measure of impulsivity.

## 5CSRTT testing: Distraction paradigm

The Distraction paradigm altered the onset of a distractive auditory stimulus presentation after trial initiation. Specifically, a distracting noise of 105dB was presented either at 0.0, 0.5,

# Testing - Long Delay

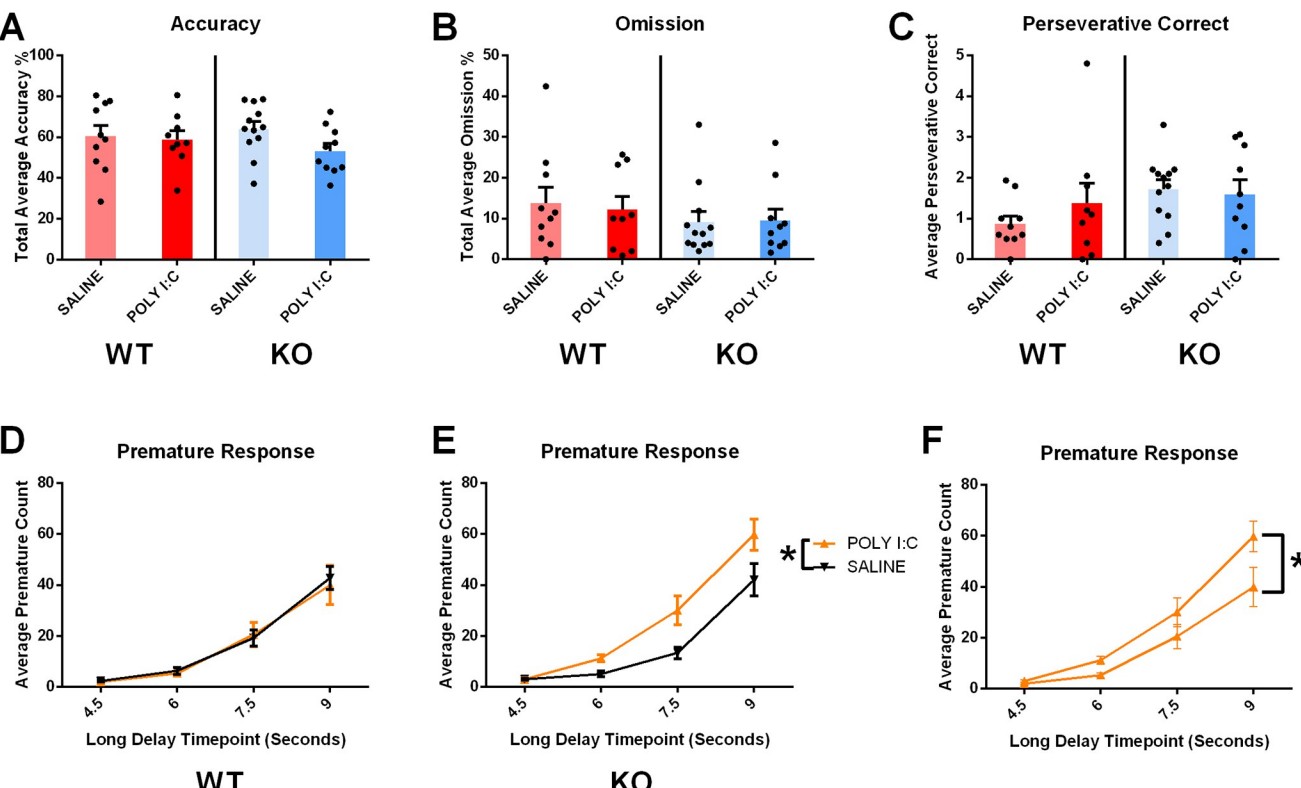

**Fig 3. Long delay 5CSRTT.** A) When tested under the long-delay paradigm, the rats did not exhibit model-dependent accuracy changes; B) or changes in omission. C) There was no significant risk factor effect on the perseverative response measure in this testing paradigm. D)WT rats prenatally exposed to poly I: C did not show a significant increase in premature responses compared to their saline counterpart. E) However, *Cntnap2* KO rats prenatally exposed to poly I: C showed an increase in premature responses compared to their saline counterpart. F) Poly I:C treated *Cntnap2* KO rats exhibited higher premature responses than poly I:C treated WT rats. *p < .05. Results are shown as mean ± SEM. $N_{WT/SALINE}$ = 10; $N_{WT/POLY\ I:C}$ = 9; $N_{KO/SALINE}$ = 12; $N_{KO/POLY\ I:C}$ = 10. For detailed data on different short delay intervals, please see S3 Fig in S1 File.

2.5, 4.5 or 5.0 seconds after trial initiation pseudo randomly. The latency time to stimulus onset was fixed to 5.0 seconds, in contrast to the short and long delay paradigms.

During Distraction testing, neither genotype nor prenatal exposure significantly influenced performance accuracy (genotype main effect: F(1, 37) = 0.284, p = 0.597; prenatal exposure main effect: F(1, 37) = 2.389, p = 0.131; **Fig 4A**). There was also no three-way interaction effect between genotype, prenatal exposure and distraction timepoint (F(3, 111) = 1.533, p = 0.195). However, *Cntnap2* KO rats exhibited reduced omitted trials (genotype main effect: F(1, 37) = 13.982, p = 0.001; prenatal exposure main effect: F(1, 37) = 0.835, p = 0.367; **Fig 4B**) with no three-way interaction effect (interaction effect: F(3, 111) = 0.301, p = 0.877). Rats prenatally exposed to poly I:C exhibited significantly increased premature responses at distraction time-point 0.5 seconds, regardless of genotype (genotype main effect: F(1, 37) = 0.003 p = 0.960; pre-natal exposure*distraction effect: F(3, 111) = 4.506, p = .002; prenatal exposure simple main effect: mean difference [3.271 +/- 1.324], p = 0.018; **Fig 4C**). There was no significant three-way interaction between both risk factors at any distraction level (F(3, 111) = 0.785, p = 0.537). As reported in other 5CSRTT testing paradigms, *Cntnap2* KO rats exhibited greater perseverative correct responses (genotype main effect: F(1, 37) = 4.504, p = 0.041; prenatal exposure

# Testing - Distraction

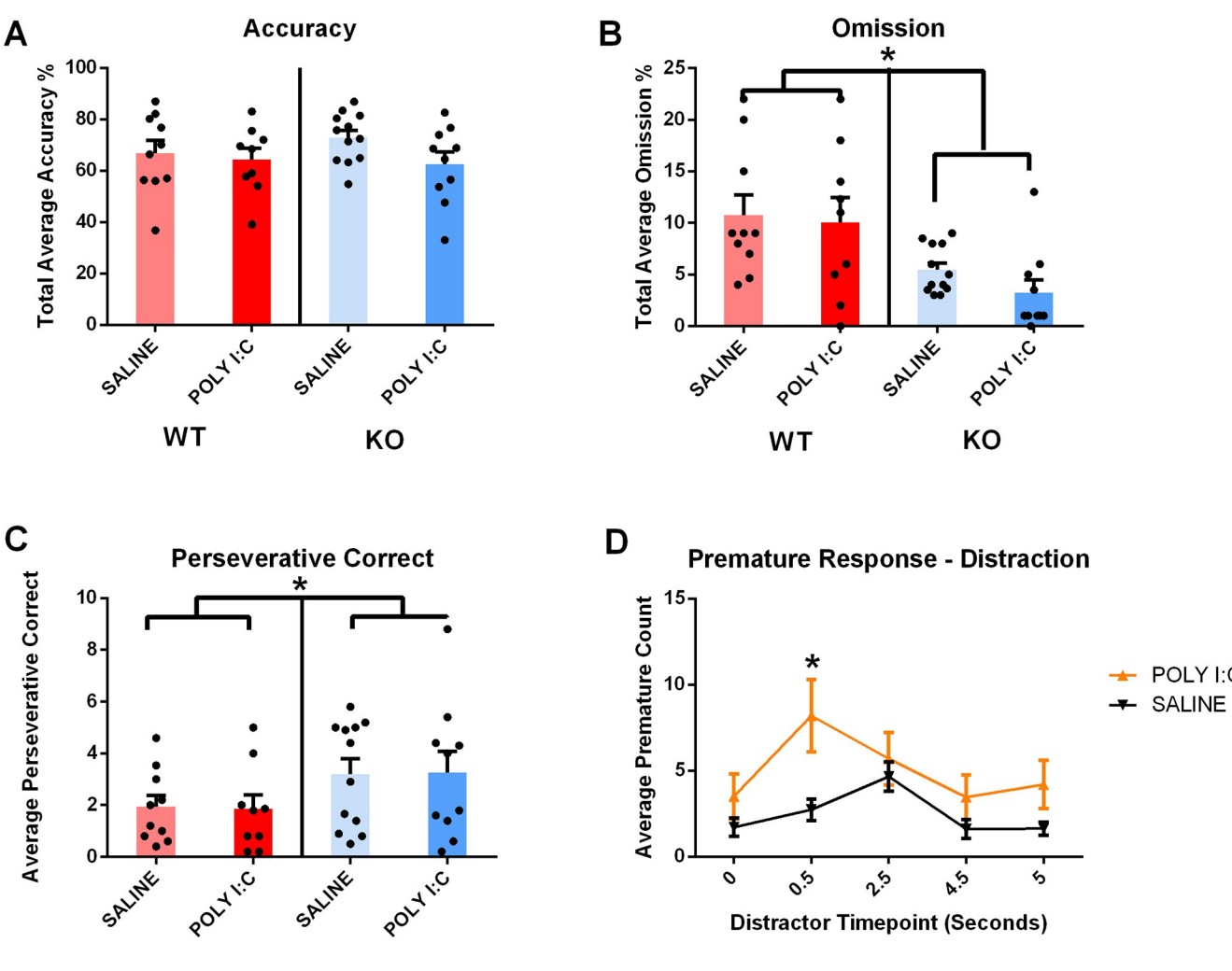

**Fig 4. Distractor 5CSRTT.** A) When tested under the Distractor paradigm, the rats did not exhibit model-dependent accuracy changes. B) Rats with *Cntnap2* deficiency exhibited lower omission rates, regardless of prenatal exposure. C) Rats with *Cntnap2* deficiency exhibited increased perseverative correct responses regardless of injection type. D) Rats prenatally exposed to poly I:C exhibited increased premature responses at a distractor timepoint of 0.5 seconds. *p < .05. Results are shown as mean ± SEM. $N_{WT/SALINE}$ = 10; $N_{WT/POLY I:C}$ = 9; $N_{KO/SALINE}$ = 12; $N_{KO/POLY I:C}$ = 10. For detailed data on different short delay intervals, please see S4 Fig in S1 File.

main effect: F(1, 37) = 0.001, p = 0.977; **Fig 4D**), while there were no three-way interaction effects (interaction effect: F(3, 111) = 0.437, p = 0.781).

The Distractor 5CSRRT paradigm revealed a lower rate of omissions in *Cntnap2* deficient rats, while once more confirming their higher perseveration rate. Furthermore, the premature responses in prenatally exposed poly I:C animals seemed to be enhanced in *Cntnap2* KO animals only, but this could only be observed at one specific distraction timepoint.

## Discussion

This study explored the separate and combined effects of *Cntnap2* deficiency and poly I:C maternal immune activation on relevant measures of attentional processing. The research

aimed to better assess the impact of genetic-environment interaction on ASD associated impairments of cognitive function, using the 5-CSRTT. The 5-CSRTT quantifies attention, impulsivity, and cognitive flexibility in preclinical models for ASD. Several previous studies have shown the construct validity of the 5-CSRTT as a model for attention [41, 44]. Due to the equalized effects of genetic and environmental risk factors, it was initially hypothesized that *Cntnap2* deficiency would interact with poly I:C MIA to exacerbate ASD-related attentional alterations seen in each model individually. However, the presented results indicated only one incidence of such an interaction across all relevant measures in the 5-CSRTT.

## Baseline training

In the baseline training stage, *Cntnap2* $^{-/-}$ and poly I:C MIA did not interact to exacerbate ASD-associated symptoms across the measures of accuracy, omission, premature and perseverative correct responses. *Cntnap2* $^{-/-}$ rats exhibited increased perseverative response in comparison to their wildtype counterpart; however, no other measures were influenced by either model alone. The baseline data raised several questions, including whether both risk factors influenced typical attentional processing during training, or if task acquisition posed a sufficient cognitive challenge to differentiate between the experimental groups.

Prior 5-CSRTT studies with separate ASD models provide varying results. In a 5-CSRTT study by Anshu and colleagues [38], valproic acid was administered prenatally at GD 12.5. The valproic acid model is consistently used in the field of ASD research due to its construct and face validity [45]. Male Sprague-Dawley rats prenatally exposed to valproic acid showed poorer performance at the first and last stages of baseline acquisition [38]. Contrastingly, a neonatal white-matter injury (WMI) study did not find a significant difference in accuracy between treatment and wildtype counterparts during baseline training. WMI can lead to hindered cognitive processing and an increased risk of ASD [46]. These disagreeing results suggest that baseline acquisition is dependent on the ASD risk-factor models' ability to learn a novel task, or different baseline protocols that can be employed in the 5-CSRTT [47]. The double-hit *Cntnap2* KO and prenatally exposed rats did not exhibit any deficit in learning and performing the task at baseline.

*Cntnap2* KO rats showed an increase in perseverative correct responses in the baseline training paradigm. Perseverative responses are defined as continuous pokes after the correct stimuli has been selected. The continuous response is a marker of compulsiveness–a tendency to perform repetitive adverse behaviors [48]. The rat's maladaptive tendency to continually nose poke a previously rewarded aperture can also act as a marker of cognitive inflexibility [49].

## Variable short and long delay

The delay and distraction paradigms highlighted key differences between treatment groups in the test stage. Firstly, decreased accuracy in poly I:C exposed rats was seen at the shortest delay value alone. There are no direct models of ASD that display variable Short-Delay outcomes; however, prenatal exposure to poly I:C can lead to cognitive inflexibility [50]. Rats prenatally exposed to poly I:C show persistence in latent inhibition (LI), a process by which introduction to a nonreinforced stimulus affects the subsequent learning of a matching reinforced stimulus [50]. LI is also a key process for assessing cognitive inflexibility. Although the 5-CSRTT does not assess LI, novel delays may challenge their preconceived understanding of the task protocol, requiring adaptive mechanisms to maintain high performance. Similarly, the study investigating white matter insult discovered poorer accuracy performance with a variable intertrial interval [46].

In the short-delay paradigm, the *Cntnap2* KO rats had a lower omission rate than their wildtype counterparts; however, this behavioral difference was only apparent in non-poly I:C exposed rats. *Cntnap2* KO rats prenatally exposed to poly I:C did not have significantly lower omission rates in comparison to WT rats also exposed to poly I:C. At the same time, *Cntnap2* KO rats showed greater perseverative correct responses in the Short-Delay paradigm [49]. Previous 5-CSRTT studies did not employ the *Cntnap2* model; nonetheless, other behavioral paradigms were used to assess repetitive behavior. A study by Wang et al. [51] found that *Cntnap2* KO mice exhibited increased grooming, a robust phenotype for repetitive behaviors. A recent paper by Scott et al. [32] identified that *Cntnap2* KO rats showed increased full body rotations and self grooming compared to the wildtype counterpart. These measures of repetitive behavior are based on instinctive and uncontrolled actions. The 5-CSRTT provides insight into repetitive behaviors that are newly developed after exposure to a novel task; nonetheless, there could be a neural mechanism that links repetitive and perseverative actions across behavioral paradigms. A 5-CSRTT study that lesioned an important region in the dopaminergic system (core subregion of the nucleus accumbens) found comparable perseverative behaviors and lack of inhibitory control [52]. Remarkably, antagonizing dopaminergic receptor D2 decreased repetitive self grooming and perseverative behaviors in *Cntnap2* KO mice [53]. Consequently, dopaminergic signalling may play a large role in modulating repetitive behaviors seen in instinctive and learned behaviors [53].

The Long-Delay paradigm elicited an interactive effect for premature responses between genotype and MIA. To elaborate, *Cntnap2* $^{-/-}$ rats exposed to MIA had a greater number of premature responses to their saline treated counterparts. Contrarily, wildtype rats exposed to MIA did not have greater premature tendencies compared to the saline treated wildtypes. Premature responses are a staple measure of impulsivity in preclinical rodent models [54, 55]. Impulsivity arises as a phenotype for atypical inhibitory response control [56]. Additionally, the rats may have exhibited impulsive tendencies due to a phenomenon known as delay-discounting [57]. With the increased stimulus delay, the value of the reward can depreciate. Therefore, the rat prematurely prompts the screen as opposed to waiting for the stimulus. Delay-discounting relies on the dorsolateral prefrontal cortex, a region involved in attention and inhibitory control [58].

## Distraction effect on 5-CSRTT performance

The distraction paradigm elicited three separate parameter effects. A distractor was played pseudo randomly at four different timepoints preceding the stimulus onset. Similarly to the short-delay paradigm, *Cntnap2* $^{-/-}$ rats had less omissions than their wildtype counterparts. This finding is not consistent across previous ASD 5-CSRTT studies [38, 55]; however, the study that looked at WMI found a decrease in omission rate with a fixed visual distractor [46]. Despite this, the present study utilized an auditory distractor. The rise in premature and perseverative responses may act as an alternative explanation in the distraction protocol. Due to increased interaction with the screen, the rats are less likely to omit a trial entirely.

Inhibitory response control and cognitive inflexibility can explain the rise in perseverative and premature responses. Like the short delay paradigm, the rats performed repetitive behaviors as a maladaptive response to a novel stimulus. On the other hand, the premature responses present a novel outcome in the distractor task. Different from the Long-Delay paradigm, there is no interactive effect between genotype and MIA. Rather, poly I:C MIA triggered premature responses at the earliest distractor timepoint alone. Poly I:C may potentially affect an inhibitory response mechanism that *Cntnap2* $^{-/-}$ did not. Rodents utilize temporal strategies to 'time' when they should attend to the oncoming stimulus [55]. It is a mediating behavior developed

because of training. The distractor can interfere with the rat's temporal strategy, triggering a premature response [55]. The dorsolateral prefrontal cortex (DLPFC) is involved in both temporal strategies, and sensory hypersensitivity to increased stimuli [59, 60]. Poly I:C has been found to increase dendritic branching in the pyramidal cells of the DLPFC, potentially explaining an increase in impulsivity due to an auditory distractor [61].

## Limitations

Although the findings provide insight into the attentional processing of a gene-environment model, future studies must address respective changes in neurobiological function to better understand the behavioral phenotype. One potential neurophysiological confound to our findings is motor planning and execution. The prefrontal cortex is known to have an inhibiting modulatory control over the premotor cortex, the area most associated with motor planning [62]. In fact, several studies on this cortico-cortical connection state that altered prefrontal activity results in motor impulsivity [63, 64]. Specifically, response cells in the premotor cortex may fire prematurely because of poor prefrontal cortex inhibition [63]. Future 5-CSRTT studies must assess if there is a significant correlation between involuntary motor impulsivity and attentional impulsivity because of premature anticipatory neuronal firing. Moreover, the 5-CSRTT task cannot act as a standalone measure of attention. For example, the continuous performance task requires the rodent to discriminate between difference stimuli based on changes in color, brightness, contrast, etc [65]. Employing additional paradigms may provide a more holistic understanding of the impact of gene-environment risk model.

## Future directions and conclusion

In conjunction with behavioral assessment, subsequent studies must look at the neurobiological explanation behind the behavioral impairments demonstrated in this study. Regarding attentional involvement of *Cntnap2* and poly I:C, a more causative link between behavioral output and mechanistic explanations can be made. Utilizing deep brain stimulation to attenuate ASD-phenotypes can directly look at the causative impact of risk factors. A study by Bekovsky et al. [66], found that the use of deep brain stimulation attenuated ASD-like sensorimotor and latent inhibition dysfunction in a poly I:C rat model. Potentially using the targets such as the dorsolateral prefrontal cortex and the ventral striatum, we can better appreciate the causal link behind atypical brain development and atypical attentional processing.

The present investigation showed that both MIA and genetic knockout can affect the cognitive process of attention separately; The *Cntnap2* KO model showed conserved changes in repetitive behavior, while the MIA model exacerbated impulsivity and lack of inhibitory control. Although neural alterations may converge across both models, a double-hit model does not certainly result in additive behavioral effects. However, attention deficits can be exacerbated in the double-hit model when presented with specific external stimuli, e.g., an auditory distractor. The parameters of this touchscreen task may also negate certain environmental influences that are generally present in a more natural environment. Due to the heterogeneity of ASD phenotypes, continually characterizing double-hit models will facilitate our understanding about the etiological basis of the disorder.

## Supporting information

**S1 File.**
(DOCX)

## Author Contributions

**Conceptualization:** Faraj Haddad, Brian Allman, Susanne Schmid.

**Data curation:** Melvin Maroon, Ella Doornaert.

**Funding acquisition:** Susanne Schmid.

**Investigation:** Melvin Maroon, Ella Doornaert.

**Methodology:** Melvin Maroon, Susanne Schmid.

**Project administration:** Susanne Schmid.

**Resources:** Susanne Schmid.

**Supervision:** Melvin Maroon, Faraj Haddad, Brian Allman, Susanne Schmid.

**Validation:** Susanne Schmid.

**Visualization:** Melvin Maroon.

**Writing – original draft:** Melvin Maroon.

**Writing – review & editing:** Faraj Haddad, Susanne Schmid.

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
