## [Decision Letter · Decision Letter 0]

7 Jul 2023

PONE-D-23-14064Investigating Gene-Environment Interaction on attention in a Double-Hit Model for Autism Spectrum DisorderPLOS ONE

Dear Dr. Schmid,

Thank you for submitting your manuscript to PLOS ONE. After careful consideration, we feel that it has merit but does not fully meet PLOS ONE’s publication criteria as it currently stands. Therefore, we invite you to submit a revised version of the manuscript that addresses the points raised during the review process.

We look forward to receiving your revised manuscript.

Kind regards,

Luca Aquili

Academic Editor

PLOS ONE

Journal Requirements:

Reviewers' comments:

Reviewer's Responses to Questions

**Comments to the Author**

1. Is the manuscript technically sound, and do the data support the conclusions?

Reviewer #1: Partly

Reviewer #2: Partly

Reviewer #3: Partly

2. Has the statistical analysis been performed appropriately and rigorously? 

Reviewer #1: Yes

Reviewer #2: Yes

Reviewer #3: I Don't Know

3. Have the authors made all data underlying the findings in their manuscript fully available?

Reviewer #1: Yes

Reviewer #2: Yes

Reviewer #3: Yes

4. Is the manuscript presented in an intelligible fashion and written in standard English?

Reviewer #1: Yes

Reviewer #2: Yes

Reviewer #3: No

5. Review Comments to the Author

Reviewer #1: In this study, the authors investigated the effects of double hit impact of genetic and environmental risk factors on attention and impulsivity related behaviors in rats using 5-CSRT paradigm. It is interesting topic to examine considering the extremely complex etiological factors associated with manifestation and pathophysiology of ASD. Not only double but also multiple including triple hits are gaining more attention in this field. Initial expectation in this filed was that combining multiple etiological factors may induce greater phenotypes in experimental animals but there is also a growing acknowledgement that multiple hits may be mutually antagonistic in some measures depending on the modality. This is understandable considering sometimes completely different direction of neurobiological changes even though all the models produce ASD-like behavioral phenotypes. This is why the investigation of underlying neurobiological changes is important in this type of study. Unfortunately, the nature of the present investigation is rather preliminary. Social and other repetitive behavioral measures need to be performed. With the inclusion of heterozygotes animal in the study or with the suboptimal concentration of pI:C, the authors may gain better insights into the nature of gene and environmental interaction. I hope I can see those further studies in the near future.

Reviewer #2: This is an interesting paper on the role of the ASD-linked gene Cntnap2 and maternal immune activation (MIA) on behaviour in the 5-choice serial reaction time task (5CSRTT). I have several suggestions for improvements to the presentation of this study.

1. Abstract: Twice you say results were "seen across two test paradigms" but do not explain what these paradigms are. This should be explained further, or these statements could even be removed from the abstract without losing any important information if there is not enough space to explain.

2. Materials and Methods- Animals: Were the WT and KO animals used in the study littermates from the heterozygous crossings? Where you say "Prior to PD 100, the offspring were placed in . . ." I think it would make more sense to say "From weaning (at XX age), the offspring were placed in . . .", then in the next section say what happened from PD 100 as you have. Some additional information would also be useful: Were rats group housed or single housed at any stage of the project? Was touchscreen testing conducted during the dark or light phase of the light cycle?

3. Materials and Methods- 5CSRTT: You have said that once the rats completed pretraining, they had 40 days to complete the training phase prior to testing. Did any rats not complete the training in 40 days or less? If so, what happened with these rats?

4. Methods/Results: If the rats had to achieve greater than 80% accuracy, and omit less than 20% of trials to move on to the next step at each stage of training, why is the accuracy in most of your results, in particular for Baseline 13, but also for the test manipulations below 80%? In terms of the stats, did you perform outlier analyses on any of the data?

5. Results: The statistics presented for each of the manipulation test session is missing a lot of details. For example, in the Short Delay section for the attention stats, you say "There was no interaction between genotype and injection type at any delay" and give just one F and p value. Is this the data for the genotype*injection type*delay timepoint interaction? In addition, it would be good to present whether there is a main effect of delay/distraction on any of the factors, just to show how the tests work in general. In all cases you should report the main effect of genotype, poly I:C and delay/distraction level, as well as genotype x poly I:C interaction and also genotype x delay/distraction, poly I:C x delay/distraction and genotype x poly I:C x delay/distraction interactions. In cases where you do seen an interaction, what did you do next with the data? For example, once you found a poly IC x distraction interaction, did you then look at the main effect of poly IC for each distraction time separately? What were the p values for this? I realise this may make the results very long, so the full results could be included as supplementary material, with the most relevant interactions only included in the main manuscript. At the very least you need to be more clear what interactions you are presenting stats for throughout the results.

You should also be consistent in the terminology you use throughout the results for each of your factors. E.g. use either MIA or poly I:C, not injection type, but choose just one of these options and stay consistent. Also, I think delay is easier to understand than timepoint for the short and long delay tests. None of the probe test trials really have a timepoint factor.

6. Figures: I would like to see the data split by delay like you have done for Short Delay-Accuracy for all measures. Whether this is in the main manuscript or supplementary material is up to you. It should be made clear in the figure legend that the figures as most are presented now include the values for all delays/distractions averaged together, and explain why you have presented it like this (e.g. if there is no significant interaction between delay/distraction and either genotype, poly I:C or both). Make the significance * bigger and clearer on all of your figures that have this included.

In figure 2C the poly I:C KO bar doesn't look right, as all the individual values are well below the mean. Is there an outlier that has been left out of the figure with the scale used (>5)?

7. Discussion: Did you compare number of sessions each group took to pass each stage of training as a measure of learning? This could be important for your baseline training and accuracy section of the discussion. As you are training the mice to reach a certain level of accuracy and omissions at each training step, these measures should not be different at baseline, although premature and perseverative responses could be, as could the number of sessions to reach baseline, and would be good to discuss in this section. The final sentence of this section doesn't really make sense.

8. Discussion: In the variable short and long delay section you have said "The Cntnap2 deficient rats similarly showed increased perseverative responses in the final baseline stage, highlighting the cognitive challenge of a shortened delay period on performance." I'm not sure how the same result in baseline and the probe tests (short delay, distraction, trend looks similar in long delay also) highlights this. It just shows something overall about the KO rats. This is also an interesting result but should be discussed differently.

9. Discussion: The discussion overall is quite disjointed with sentences and examples from the literature that don't link together well and need much more explanation about their relevance to results of the current study.

Also, the links to ASD and your introduction are not strong and need more work to make the relevance of this study clear. This is the section that needs the most work to make it suitable for publication. All co-authors should read through this section closely to make improvements.

9. Conclusion: This section doesn't really make any conclusions about the current study.

Reviewer #3: The study by Maroon and coworkers aims to investigate the effects of maternal immune activation in combination with Cntnap 2 deficiency on attentional capabilities assessed in 5CSRT task.

I have several concerns as follows.

1.The title is too generic.

2.Page 5 lines 133-134 Were saline- or Poly I:C-treated dams heterozygous? How many dams for treatment were used?

Moreover, authors assessed only males. It’s a pity that authors did not assess also female rats because it would be interesting to investigate potential sex differences in the vulnerability to double hits.

3.Authors should review all the manuscript: there are many inaccuracies, for example: - Poly I:C or poly I:C?;

- injection type: it would be better to use the term “prenatal exposure/administration”, and so on.

4. Authors should better round out the description of results. For example, it would be more correct to write "genotype" and "prenatal administration" effects or "Ctnap2 deficiency" and "Poly I:C".

6. PLOS authors have the option to publish the peer review history of their article (what does this mean?). If published, this will include your full peer review and any attached files.

Reviewer #1: No

Reviewer #2: No

Reviewer #3: No

---

## [Author Response · Author response to Decision Letter 0]

7 Sep 2023

Review Comments to the Author

Reviewer #1:

 In this study, the authors investigated the effects of double hit impact of genetic and environmental risk factors on attention and impulsivity related behaviors in rats using 5-CSRT paradigm. It is interesting topic to examine considering the extremely complex etiological factors associated with manifestation and pathophysiology of ASD. Not only double but also multiple including triple hits are gaining more attention in this field. Initial expectation in this filed was that combining multiple etiological factors may induce greater phenotypes in experimental animals but there is also a growing acknowledgement that multiple hits may be mutually antagonistic in some measures depending on the modality. This is understandable considering sometimes completely different direction of neurobiological changes even though all the models produce ASD-like behavioral phenotypes. This is why the investigation of underlying neurobiological changes is important in this type of study. Unfortunately, the nature of the present investigation is rather preliminary. Social and other repetitive behavioral measures need to be performed. With the inclusion of heterozygotes animal in the study or with the suboptimal concentration of pI:C, the authors may gain better insights into the nature of gene and environmental interaction. I hope I can see those further studies in the near future.

Reviewer #2: 

This is an interesting paper on the role of the ASD-linked gene Cntnap2 and maternal immune activation (MIA) on behaviour in the 5-choice serial reaction time task (5CSRTT). I have several suggestions for improvements to the presentation of this study.

1. Abstract: Twice you say results were "seen across two test paradigms" but do not explain what these paradigms are. This should be explained further, or these statements could even be removed from the abstract without losing any important information if there is not enough space to explain.

The “across two test paradigms” phrases were removed as per the suggestion. To expand on them would affect the concision of the abstract, but their presence does affect the clarity of the summary. 

2. Materials and Methods- Animals: Were the WT and KO animals used in the study littermates from the heterozygous crossings? 

Yes, both genotypes were obtained from heterozygous crossings: “The n-value may differ in the displayed test results as certain rats were seizure prone and exhibited atypical baseline behavior. Horizon Discovery (Boyertown, PA; originated at SAGE Laboratories, Inc. with Autism Speaks; the line is presently upheld by Envigo) provided heterozygous breeders with a five base-pair deletion at exon six in the Cntnap2 gene, created by zinc-finger nuclease target site CAGCATTTCCGCACC|aatgga|GAGTTTGACTACCTG. All genotypes tested in this experiment are littermates acquired from heterozygous crossings.”

Where you say "Prior to PD 100, the offspring were placed in . . ." I think it would make more sense to say "From weaning (at XX age), the offspring were placed in . . .", then in the next section say what happened from PD 100 as you have. Some additional information would also be useful: Were rats group housed or single housed at any stage of the project? Was touchscreen testing conducted during the dark or light phase of the light cycle?

The change was made to “From weaning ()”. The additional information was also added to the methodology section including housing situation and that the rats were only tested during the light-cycle. 

3. Materials and Methods- 5CSRTT: You have said that once the rats completed pretraining, they had 40 days to complete the training phase prior to testing. Did any rats not complete the training in 40 days or less? If so, what happened with these rats?

If any remaining rats did not reach the final baseline level by the end of the training stage, they were moved to baseline 13 alongside the rats passed within the 40 days and were run on the final training stage. This was done to avoid overtraining. Any outliers from the rats moved up (using boxplots and interquartile ranges) were removed prior to the testing protocols. This is now highlighted at the end of the training section of the methodology. 

4. Methods/Results: If the rats had to achieve greater than 80% accuracy, and omit less than 20% of trials to move on to the next step at each stage of training, why is the accuracy in most of your results, in particular for Baseline 13, but also for the test manipulations below 80%? In terms of the stats, did you perform outlier analyses on any of the data?

An outlier analysis was run at baseline, any rats considered extreme outliers were removed prior to moving on to the probing sessions. The performance rose above 80% leading to the final baseline training stage to pass preceding levels. Due to the very short stimulus duration of baseline 13, most of the rats performed below the 80% even by the end of the training period. We ran rats for longer than 40 days in a pilot cohort to see if they would improve; however, in some cases performance even declined if they were overtrained. Therefore, a number of rats did not achieve 80% accuracy on the last training stage and extreme outliers were removed to not skew and misrepresent the data. The probe sessions, Short/Long Delay, and the Distraction protocol provide a novel challenge to the rats which would affect their behavior and decrease their accuracy performance further. As they are not used to adjusted parameters, it is expected to see a decline in accuracy compared to baseline and easier training levels. 

5. Results: The statistics presented for each of the manipulation test session is missing a lot of details. For example, in the Short Delay section for the attention stats, you say "There was no interaction between genotype and injection type at any delay" and give just one F and p value. Is this the data for the genotype*injection type*delay timepoint interaction? 

Yes, it is the p-value for the three-way interaction as a repeated measures analysis was run. As there was no significance for the three-way, that would indicate that there is no effect of either genotype or treatment on any of the individual levels of the within-subjects factor. This was clarified by including “three-way” in the result section. 

In addition, it would be good to present whether there is a main effect of delay/distraction on any of the factors, just to show how the tests work in general. In all cases you should report the main effect of genotype, poly I:C and delay/distraction level, as well as genotype x poly I:C interaction and also genotype x delay/distraction, poly I:C x delay/distraction and genotype x poly I:C x delay/distraction interactions. In cases where you do seen an interaction, what did you do next with the data? For example, once you found a poly IC x distraction interaction, did you then look at the main effect of poly IC for each distraction time separately? What were the p values for this? I realise this may make the results very long, so the full results could be included as supplementary material, with the most relevant interactions only included in the main manuscript. At the very least you need to be more clear what interactions you are presenting stats for throughout the results.

When running the statistical analyses, when a three or two-way interaction with the within-subjects factor was found, follow-up pairwise comparison post-hoc tests were done to find the simple main effects. The simple main effects were clarified in the discussion, so the direction of the findings is more evident. 

You should also be consistent in the terminology you use throughout the results for each of your factors. E.g. use either MIA or poly I:C, not injection type, but choose just one of these options and stay consistent. Also, I think delay is easier to understand than timepoint for the short and long delay tests. None of the probe test trials really have a timepoint factor.

The terminology was made more consistent throughout the manuscript. There are only changes in terminology now when a specific factor needs to be discussed so an overarching term such as MIA would not suffice. Timepoint was replaced with delay level or delay timepoint. 

6. Figures: I would like to see the data split by delay like you have done for Short Delay-Accuracy for all measures. Whether this is in the main manuscript or supplementary material is up to you. It should be made clear in the figure legend that the figures as most are presented now include the values for all delays/distractions averaged together, and explain why you have presented it like this (e.g. if there is no significant interaction between delay/distraction and either genotype, poly I:C or both). Make the significance * bigger and clearer on all of your figures that have this included.

The significance asterisk was amplified from font size 14-20 to appear clearer on all figures. 

To present all data separated into each within-subjects delay level may deter from the key findings when not significant. We only combined within-subjects levels when there was no significance found across each level. Separating it would not necessarily add to the story as each level itself is insignificant in those cases. 

In figure 2C the poly I:C KO bar doesn't look right, as all the individual values are well below the mean. Is there an outlier that has been left out of the figure with the scale used (>5)?

The scaling has been adjusted to include all individual points on the graph that were previously not seen due to the automatic left y-axis boundary set by GraphPad. The mean value should make sense now with the new scaling. 

7. Discussion: Did you compare number of sessions each group took to pass each stage of training as a measure of learning? This could be important for your baseline training and accuracy section of the discussion. As you are training the mice to reach a certain level of accuracy and omissions at each training step, these measures should not be different at baseline, although premature and perseverative responses could be, as could the number of sessions to reach baseline, and would be good to discuss in this section. The final sentence of this section doesn't really make sense.

The number of days to reach final baseline criterion was analyzed using a univariate general linear model analysis. There was no significant interaction between genotype and prenatal exposure, so it was not included in the present paper. The aim was more centrally focused on attentional parameters, and since there was no effect on learning regarding days to criterion, this was no shown in the manuscript. 

8. Discussion: In the variable short and long delay section you have said "The Cntnap2 deficient rats similarly showed increased perseverative responses in the final baseline stage, highlighting the cognitive challenge of a shortened delay period on performance." I'm not sure how the same result in baseline and the probe tests (short delay, distraction, trend looks similar in long delay also) highlights this. It just shows something overall about the KO rats. This is also an interesting result but should be discussed differently.

This part was amended accordingly. Rather than compare baseline and short delay repetitive behavior, the basis behind repetitive behaviors was more deeply discussed so as to avoid any confusion in comparing different section outcomes. 

9. Discussion: The discussion overall is quite disjointed with sentences and examples from the literature that don't link together well and need much more explanation about their relevance to results of the current study.

Also, the links to ASD and your introduction are not strong and need more work to make the relevance of this study clear. This is the section that needs the most work to make it suitable for publication. All co-authors should read through this section closely to make improvements.

9. Conclusion: This section doesn't really make any conclusions about the current study.

Adjustments were made to try to clarify the discussion better. Some sections were rearranged such as the baseline and delay, with some previously unclear statements elaborated on. The conclusion was also amended to be more definitive. 

Reviewer #3: The study by Maroon and coworkers aims to investigate the effects of maternal immune activation in combination with Cntnap 2 deficiency on attentional capabilities assessed in 5CSRT task.

I have several concerns as follows.

1.The title is too generic.

Title was changed to “Assessing visuospatial attention in a Gene-Environment Model of Cntnap2 Knockout and Poly I:C Maternal Immune Activation”

2.Page 5 lines 133-134 Were saline- or Poly I:C-treated dams heterozygous? How many dams for treatment were used?

16 dams were treated (n=8 poly I:C and n= 9 saline)

Yes, both genotypes were obtained from heterozygous crossings: “The n-value may differ in the displayed test results as certain rats were seizure prone and exhibited atypical baseline behavior. Horizon Discovery (Boyertown, PA; originated at SAGE Laboratories, Inc. with Autism Speaks; the line is presently upheld by Envigo) provided heterozygous breeders with a five base-pair deletion at exon six in the Cntnap2 gene, created by zinc-finger nuclease target site CAGCATTTCCGCACC|aatgga|GAGTTTGACTACCTG. All genotypes tested in this experiment are littermates acquired from heterozygous crossings.”

See also comments above.

Moreover, authors assessed only males. It’s a pity that authors did not assess also female rats because it would be interesting to investigate potential sex differences in the vulnerability to double hits.

This point is certainly extremely valid, this investigation was a continuation of behavioral assessments conducted between PD0-100, therefore the rats used were all males. Males often show a more severe phenotype, so this first study was conducted with males. However, for future investigations it is certainly imperative to investigate sex effects and even look at the effects of the estrous cycle on behavioral outcomes. 

3.Authors should review all the manuscript: there are many inaccuracies, for example: - Poly I:C or poly I:C?;

Inaccuracies were adjusted. 

- injection type: it would be better to use the term “prenatal exposure/administration”, and so on.

This was changed, see also comments to reviewer #1

4. Authors should better round out the description of results. For example, it would be more correct to write "genotype" and "prenatal administration" effects or "Ctnap2 deficiency" and "Poly I:C".

Abbreviations were made consistent throughout the paper, and injection type was replaced with prenatal exposure through the results section. Cntnap2 deficiency was used instead of -/- and MIA was used to provide more clarity to the results section. “Prenatal poly I:C exposure” was used when interactions or a particular finding was seen.

---

## [Decision Letter · Decision Letter 1]

24 Oct 2023

PONE-D-23-14064R1Assessing visuospatial attention in a Gene-Environment Model of Cntnap2 Knockout and Poly I:C Maternal Immune ActivationPLOS ONE

Dear Dr. Schmid,

Thank you for submitting your manuscript to PLOS ONE. After careful consideration, we feel that it has merit but does not fully meet PLOS ONE’s publication criteria as it currently stands. Therefore, we invite you to submit a revised version of the manuscript that addresses the points raised during the review process (please see section 6 of this email).

We look forward to receiving your revised manuscript.

Kind regards,

Luca Aquili

Academic Editor

PLOS ONE

Journal Requirements:

Reviewers' comments:

Reviewer's Responses to Questions

**Comments to the Author**

1. If the authors have adequately addressed your comments raised in a previous round of review and you feel that this manuscript is now acceptable for publication, you may indicate that here to bypass the “Comments to the Author” section, enter your conflict of interest statement in the “Confidential to Editor” section, and submit your "Accept" recommendation.

Reviewer #2: (No Response)

2. Is the manuscript technically sound, and do the data support the conclusions?

Reviewer #2: Yes

3. Has the statistical analysis been performed appropriately and rigorously? 

Reviewer #2: Yes

4. Have the authors made all data underlying the findings in their manuscript fully available?

Reviewer #2: Yes

5. Is the manuscript presented in an intelligible fashion and written in standard English?

Reviewer #2: Yes

6. Review Comments to the Author

Reviewer #2: I thank the authors for responding to most of my comments, that manuscript is improved and mush easier to follow. There are some additional results I would still like to see in a supplementary results file, which would not deter from the key

findings as you were concerned about in the main manuscript. These include:

1. It would be good to present whether there is a main effect of delay/distraction on any of the factors, just to show how the tests work in general. You have only presented the main effects of genotype and prenatal exposure. When using a repeated measures ANOVA with different within subjects factors of delay or distraction there should be a main effect of these, for example it appears there would be a main effect of increasing delay on premature responses in the long delay test.

For completeness it would be good to list the 2 way interactions as well in supplementary results, i.e. genotype x delay/distraction, poly I:C x delay/distraction and genotype x poly I:C interactions. This could just be in a large table for simplicity.

2. Figures: I would like to see the data split by delay like you have done for Short Delay-Accuracy for all measures in a supplementary file. I agree this would be too much for the main manuscript, however it should be made clear in the figure legends of the main manuscript that the figures include the values for all delays/distractions averaged together, and explain why you have presented it like this (e.g. if there is no significant interaction between delay/distraction and either genotype, poly I:C or both).

3. The data for number of days to reach criterion which you said you have analysed would be good in a supplementary results file as well.

7. PLOS authors have the option to publish the peer review history of their article (what does this mean?). If published, this will include your full peer review and any attached files.

Reviewer #2: No

---

## [Author Response · Author response to Decision Letter 1]

16 Jan 2024

Reviewer #2: I thank the authors for responding to most of my comments, that manuscript is improved and mush easier to follow. There are some additional results I would still like to see in a supplementary results file, which would not deter from the key findings as you were concerned about in the main manuscript. These include:

1. It would be good to present whether there is a main effect of delay/distraction on any of the factors, just to show how the tests work in general. You have only presented the main effects of genotype and prenatal exposure. When using a repeated measures ANOVA with different within subjects factors of delay or distraction there should be a main effect of these, for example it appears there would be a main effect of increasing delay on premature responses in the long delay test.

For completeness it would be good to list the 2 way interactions as well in supplementary results, i.e. genotype x delay/distraction, poly I:C x delay/distraction and genotype x poly I:C interactions. This could just be in a large table for simplicity.

We have created a table in the supplementary file showing the two- and three-way interactions with the within-subject factors across the three testing paradigms, as requested. 

We also want to clarify an error caught in the manuscript. We reran all the analyses when asked to parse the data for the supplementary figures. All data was properly replicated; however, an error was caught in the omission data for the short-delay paradigm. We correctly listed the new values and amended the discussion accordingly to incorporate the significant findings. We apologize for this error and are thankful that it was caught before the paper was finalized. 

2. Figures: I would like to see the data split by delay like you have done for Short Delay-Accuracy for all measures in a supplementary file. I agree this would be too much for the main manuscript, however it should be made clear in the figure legends of the main manuscript that the figures include the values for all delays/distractions averaged together, and explain why you have presented it like this (e.g. if there is no significant interaction between delay/distraction and either genotype, poly I:C or both).

We split the data in the supplementary figures for those that were not already parsed in the main manuscript. If they had an interaction effect, we showed that across the different delays/distractions; if there only was a main effect, we collapsed this between-subject factor across the delays/distractions; If there were no significant effects we show a line graph for all four groups to highlight the general trend of each group.

3. The data for number of days to reach criterion which you said you have analysed would be good in a supplementary results file as well.

We added the bar graph that showed the days to criterion for each group. It is appended at the bottom of the supplementary file.

---

## [Editor Report · Decision Letter 2]

9 Feb 2024

Assessing visuospatial attention in a Gene-Environment Model of Cntnap2 Knockout and Poly I:C Maternal Immune Activation

PONE-D-23-14064R2

Dear Dr. Schmid,

We’re pleased to inform you that your manuscript has been judged scientifically suitable for publication and will be formally accepted for publication once it meets all outstanding technical requirements.

Kind regards,

Luca Aquili

Academic Editor

PLOS ONE
---

## [Editor Report · Acceptance letter]

20 Mar 2024

PONE-D-23-14064R2 

PLOS ONE

Dear Dr. Schmid, 

I'm pleased to inform you that your manuscript has been deemed suitable for publication in PLOS ONE. Congratulations! Your manuscript is now being handed over to our production team.

Kind regards, 

on behalf of

Dr. Luca Aquili 

Academic Editor

PLOS ONE